# Evaluation of the Volatile Composition and Sensory Behavior of Habanero Pepper during Lactic Acid Fermentation by *L. plantarum*

**DOI:** 10.3390/foods11223618

**Published:** 2022-11-12

**Authors:** Diego López-Salas, Julio Enrique Oney-Montalvo, Emmanuel Ramírez-Rivera, Manuel Octavio Ramírez-Sucre, Ingrid Mayanin Rodríguez-Buenfil

**Affiliations:** 1Sede Sureste CIATEJ, Centro de Investigación y Asistencia en Tecnología y Diseño del Estado de Jalisco A.C., Tablaje Catastral 31264, Carretera Sierra Papacal-Chuburna Puerto km. 5.5, Parque Científico Tecnológico de Yucatán, Mérida 97302, Mexico; 2Departamento de Innovación Agrícola Sustentable, Tecnológico Nacional de México/Tecnológico Superior de Zongolica, Carretera S/N km. 4, Tepetlitlanapa, Zongolica 95005, Mexico

**Keywords:** *Capsicum chinense*, Habanero pepper, lactic acid bacteria, sensory analysis, volatile compounds, emotions

## Abstract

Habanero pepper is recognized for its appealing aroma and flavor. Lactic acid fermentation can improve these sensory properties, especially aroma, by the synthesis of volatile compounds, which might also increase the consumer preference. Thus, the aim of this research was to compare the volatile composition as well as different sensory parameters such as preference and emotions related to the lactic acid fermentation of Habanero pepper by two strains (wild and commercial) of *Lactiplantibacillus plantarum*. A multiple factor ANOVA was used to compare the volatile composition with different fermentation times and strains. The results demonstrated that the interaction between the strain and fermentation time had significant effects on the volatile compound production that includes 1-hexanol, cis-3-hexenyl hexanoate, linalool, and 3,3 dimethyl-1-hexanol while only time influenced the production of trans-2-hexen-1-al. The wild strain (WIL) at 48 h of fermentation produced the highest concentration of 3,3 dimethyl-1-hexanol and trans-2-hexen-1-al. On the other hand, the commercial strain (COM) presented the highest concentration of 1-hexanol and cis-3-hexenyl hexanoate with a 72 h fermentation. The most preferred sample was that fermented by WIL for 48 h for the attribute of odor, while for taste, the most preferred sample was that fermented for 72 h with COM.

## 1. Introduction

Habanero pepper (*Capsicum chinense*) is an herbaceous plant that is considered one of the most important crops in the Yucatan Peninsula due it being locally, nationally, and internationally marketed in different presentations: (1) fresh, (2) powder, (3) pasta, and (3) sauces [1]. Its sensory properties such as aroma, flavor, pungency, and color awarded it its appellation of origin in 2010 [1].

Among the sensory properties, aroma is considered as one of the most appealing features and is associated with the volatile compounds that build its distinctive odor and contribute to the flavor [2]. The composition of volatile compounds in this fruit has been previously studied, finding from 53 to 102 different volatile compounds according to the level of ripeness and the method of isolation [3]. In general, the volatile fraction consists of low molecular weight compounds, these being: (1) aldehydes, (2) ketones, (3) alcohols, (4) esters, (5) terpenoids, and (6) lactones as the main classes of volatiles in Habanero pepper [4].

For the food industry, aroma is an important attribute due to the relationship with food quality and consumer acceptability [5]. Fermentation has been a preferred method to enhance the sensory characteristics due to the synthesis of volatile compounds with appealing odor notes and the transformation of other molecules that are related to odd flavors, an example of which is the bioconversion of hexanal to hexanoic acid in the fermentation of mung beans by lactic acid bacteria [6,7,8]. In fact, lactic acid bacteria (LAB) are extensively utilized in the food industry not only for their ability to metabolize several carbohydrates and their role as probiotics, but due to the effect on the sensory properties of fermented foods such as odor and taste, which can be associated with the volatile composition [9,10]. Thus, the monitoring of volatile compounds during fermentation by lactic acid bacteria is of utmost importance to estimate the sensory and nutritional quality of the final product [11].

The evaluation of sensory attributes and emotions in foods helps in the identification of characteristics that increase customer satisfaction, leading to a greater opportunity for purchase intent on their part [12]. The study previously conducted by Peralta-Cruz et al. [13] showed that there was a difference in the emotions caused by the degree of maturity of Habanero pepper (unripe or ripe) as well as in the sensory attributes to which they were related. Thus, the response in the emotions and sensory attributes of habanero pepper might be affected during the fermentation of this fruit due to the biochemical changes that the raw material will undergo in the process [14].

This research was purposedly made to evaluate the lactic acid fermentation effect on some the sensory properties that characterize Habanero pepper such as aroma and flavor as well as to determine how these changes impact the consumer acceptance.

Based on the aforementioned, in this study, the volatile composition as well as different sensory parameters such as preference and emotions were compared during the lactic acid fermentation of Habanero pepper by two strains of *Lactiplantibacillus plantarum*: wild and commercial. This will help determine which fermentation conditions best enhance the sensory value of Habanero pepper as well as to detect a relationship between certain volatile compounds and consumer preference.

## 2. Materials and Methods

### 2.1. Raw Matter: Capsicum chinense

Habanero pepper in a ripe maturity stage was purchased through a distributor in eastern Yucatan (Sabor del Mayab, Yucatán, México). Peduncles were separated from the fruit, which was then washed and sanitized by submersion for 10 min with tap water added to a commercial 0.175% (*v*/*v*) solution of ionized silver (Microdyn, Azcapotzalco, Mexico). Finally, the fruits were pureed in a blender (LI-3, International^®^, Mexico City, Mexico) and stored by freezing at −18 °C.

### 2.2. Bacterial Growth Conditions

Commercial strain code ECGC13110402 *L. plantarum* LDL (SACCO, Cadorago, Italy) and Habanero pepper isolated, *L. plantarum* YFPB1BMX (GenBank accession for partial 16S RNA gene sequence GenBank code: FJ538586.1) were the strains used.

Each strain was inoculated at 40 °C for 8 h with an initial cell count of 10^7^ cells·mL^−1^, which was determined by direct count with a Petroff–Hausser counting chamber (Horsham, USA) with an optical microscope (VTVIX-2, Olympus, Tokyo, Japan) [13], in a MRS or De Man Rogosa Sharpe nutritious broth (DIFCO, Le Pont de Claix, France); a sample was taken and centrifuged at 4700 rpm, 20 min at 4 °C (MEGAFUGE 40R, Bremen, Germany) and the resulting pellets were washed twice with NaCl at a 0.85% mass/volume concentration in order to retrieve the bacterial pellet to inoculate by 10% (*v*/*v*) a medium consisting of 60% (*w*/*v*) sterilized Habanero pepper puree (HPP), which was then incubated for 5 h at 40 °C. Sterilization was carried out at 121 °C, 15 psi, for 15 min.

### 2.3. Habanero Pepper Fermentation

The fermented HPP media were used as 10% inoculums to other HPP media prepared similarly, then incubated for 72 h at 40 °C. Four samples were taken during the first 8 h in a 2 h span and two times a day from the 24 h sample onward.

### 2.4. Biomass Quantification and Lactic Acid Measurement

#### 2.4.1. Quantification of Biomass

Direct microscopic count was used to determine the cell count [13], which was converted to dry weight with a correlation curve for each strain of *L. plantarum*, the commercial (COM) and the wild (WIL) one. The curve consisted of previously fermented samples of COM and WIL in MRS diluted in distilled water. Each dilution was centrifuged for 20 min, at 4 °C, with 4700 rpm and rinsed twice with distilled water, retrieving the pellet, then adding 5 mL of distilled water before the samples were dried at 60 °C for 16 h until constant weight on an analytical scale (Ohaus Explorer PA224C, Parsipanny, NJ, USA).

#### 2.4.2. Lactic Acid and pH Measurement

Spectrophotometry was used to determine the lactic acid concentration. A 50 μL aliquot was added to 2 mL of a FeCl_3_ 0.2% (*w*/*v*) solution, and after homogenization, the optical density was measured with a Jenway 6715 UV–Visible Spectrophotometer (Staffordshire, UK) at 390 nm. Concentrations for each sample were determined using a calibration curve from 1 to 10 g∙L^−1^ with a lactic acid standard (≥85%, Aqua solutions, Deer Park, TX, USA) [15]. The pH values were determined for each fermented sample using a multiparameter potentiometer PC2700 from Oakton Instruments (Vernon Hills, IL, USA) (Appendix A, Table A1).

### 2.5. Volatile Compound Extraction

Extraction of the samples occurred by a 1-h distillation of 50 mL aliquots from different fermentation times (0, 24, 48, and 72 h). Then, a liquid–liquid extraction was carried by mixing the distillates with 5 mL of dichloromethane and then recovering the insoluble phase. Finally, the extracts were quantified by gas chromatography after being concentrated to 1 mL [16,17]. Different strains and fermentation times were compared through a multifactor ANOVA with each volatile compound concentration as a response variable.

### 2.6. Volatile Compound Quantification by Gas Chromatography

A gas chromatographer with a flame ionization detector and a 0.25 μm thick, 0.25 mm i.d. × 60 m polyethylene glycol column was used (Trace GC, Thermo Scientific, Waltham, MA, USA). Extracts were injected in 2 μL on splitless mode. Injection temperature was 250 °C. Temperature in the oven was kept for 4 min at 40 °C, then, after a time of 110 min, it was raised to 150 °C, followed by a 6 °C min^−1^ increase of 90 °C to reach 240 °C, where it stayed for 2 min. The carrier gas was nitrogen, introduced at a 1 mL min^−1^ rate [16,17].

The comparison between the retention times of the samples with the standards led to the identification of volatile compounds, quantified using a calibration curve with a maximum concentration of 670 μg mL^−1^. The standards used were linalool (purity ≥95%), 2,3-buthanedione, limonene, 3,3-dimethyl-1-hexanol and linalool (purities ≥97%), trans-2-hexen-1-al, isoamyl isobutyrate, and cis-3-hexenyl hexanoate (purities ≥98%), hexyl-3-methyl butanoate and 1-hexanol (purities ≥99%). The manufacturer of the standards was Sigma-Aldrich in Toluca, Mexico. Chromatograms for the standards and samples are presented in Appendix B, Figure A1, Figure A2 and Figure A3.

### 2.7. Hedonic Sensory Analysis

#### 2.7.1. Preparation of the Samples

Samples from the different fermentation times (0, 24, 48 and 72 h) for each strain were centrifuged (4700 rpm, 20 min, 4 °C). The supernatant was recovered, then distributed in clean, transparent cups for odor sensory analysis. For odor evaluation, the supernatant for each sample was diluted in a 1:5 proportion and added to 40 g L^−1^ of sucrose and served in the same type of cup as the previous analysis [18]. All samples were evaluated at 25 °C.

#### 2.7.2. Sensory Evaluation

A total of 13 untrained panelists comprised of six women and seven men was selected. A hedonic scale of nine points was used, where 1 represented “exceedingly loathe” and 9, “exceedingly enjoy” [19]; a check-all-that-applies (CATA) assay was included where panelists associated their perception with a list of emotions (Table 1) [20]. For odor analysis, panelists sniffed coded samples in 30 s intervals. For taste analysis, the panelists took a sip from each sample with 60 s intervals between samples; in this time, they had milk and unsalted crackers to reduce pungent feelings and purified water to finally rinse their mouth [18].

### 2.8. Statistical Analysis

Both strains were compared with a one-way analysis of variance (ANOVA) with kinetic parameters as response variables, which were enlisted as follows: maximum dry weight production (∆X), growth rate (μ), maximum lactic acid concentration (∆P), lactic acid synthesis rate (Q_p_). Equations for μ, ∆X, Q_p_, and ∆P are shown below:μ · t = ∆X/∆t(1)
X_f_ − X_o_ = ∆X(2)
Q_p_ · t = ∆P/∆t (3)
P_f_ − P_o_ = ∆P(4)
where t is the fermentation time in h; X represents the dry weight concentration of bacteria (g L^−1^); the maximum dry weight at a given time is X_f_ (g L^−1^); the dry weight of biomass is initially X_o_ (g L^−1^); lactic acid as a product concentration is P (g L^−1^); the maximum product concentration is P_f_ (g L^−1^); the product concentration at the beginning of the fermentation is P_o_ (g L^−1^).

Both the ANOVA used to compare kinetic parameters between strains and the multiple factor ANOVA to compare volatile composition at different fermentation times and strains were performed by Statgraphics Technologies Inc. software Statgraphics Centurion XVI (the Plains, VA, USA) with a 95% confidence level. A CATA analysis was performed in a matrix dimensioned as *(J ∙ I) K*, where *J* = the different fermented samples, *I* = the 13 consumers, and *K* = the evaluated emotions to determine significant emotions with a Cochran’s Q test [19] with a threshold of 0.5 using XLSTAT (Addison, Paris, France). Emotions obtained were added to a principal component analysis (PCA) that also included volatile compounds and experimental factors, correlated by a Pearson analysis with a 95% confidence level coupled with a vectorial, 2-dimensional external preference map (PREFMAP) with a significance level of 5%, which was made using the same software [12].

## 3. Results

### 3.1. Bacterial Growth during Fermentation and Lactic Acid Production

Biomass and lactic acid produced during a 72 h fermentation are shown in Figure 1. Growth reached a maximum for commercial *L. plantarum* (COM) at 6 h of fermentation, maintaining a constant concentration until the end of fermentation; this parameter was apparently reached by wild *L. plantarum* (WIL) at a time between 8 and 24 h of fermentation, for the rest of the fermentation, this concentration remained unchanged. None of the strains showed any adaptation time. COM had the highest biomass concentration after the end of fermentation of both strains. Maximum lactic acid production for COM was reached by 24 h of fermentation, while WIL after 48 h, this last with the highest lactic acid concentration by the end.

### 3.2. Kinetic Parameter Analysis

Table 2 shows the kinetic parameters determined during the fermentation of Habanero pepper for each strain. No statistically significant differences were found between strains for parameters ∆X nor Q_p_, but there was a significant difference in μ (*p* value 0.0249) and ∆P (*p* value 0.0287), and COM presented the highest μ of 0.0680 ± 0.0025 h^−1^ over the 0.0336 ± 0.0074 h^−1^ of WIL, which means that COM grew twice as fast as WIL in Habanero pepper but obtained the same biomass concentration (∆X) at the end. WIL had the highest lactic acid production (10.81 ± 1.03 g∙L^−1^) even when their production rate (Q_p_) was not significantly different.

### 3.3. Volatile Compound Production during Fermentation

Volatile compound concentrations at different fermentation times are reported in Table 3. Concentrations of 2,3 butanedione, limonene, isoamyl isobutyrate, and hexyl-3-methyl butanoate remained unchanged throughout the fermentation for both strains while concentrations of 1-hexanol, cis-3-hexenyl hexanoate, 3,3 dimethyl-1-hexanol, and linalool increased as fermentation progressed. For trans-2-hexen-1-al, a maximum was achieved by WIL after 48 h. COM produced the highest amount of 1-hexanol, linalool, and cis-3-hexenyl hexanoate at 72 h, while WIL presented it for 3,3 dimethyl-1-hexanol as well as linalool at 48 h.

Table 4 shows the *p* values, which indicates the effect of each factor on the concentration of volatile compounds during fermentation. The concentrations of 1-hexanol, cis-3-hexenyl hexanoate, 3,3 dimethyl-1-hexanol, and linalool were significantly affected by the interaction between strain and time. Time was the only factor that influenced the trans-2-hexen-1-al concentration.

### 3.4. Sensory Evaluation

The preference map (PREFMAP) associated with the principal component analysis (PCA) regarding the samples, volatile compounds, and emotions for odor is shown in Figure 2. The PREFMAP is divided in different color regions that indicate the percentage of the panelists who preferred each sample by rating them closer to the “extremely like” mark; samples with emotions and volatile compounds close to one another in the same segment on the chart indicate a relationship between them. The sample in the region of most preference (80–100%) was WIL fermented for 48 h, which, as mentioned before, had the highest concentration of 3,3 dimethyl-1-hexanol as well as trans-2-hexen-1-al, and it is also mostly associated with positive attributes such as good, active, and happy. COM fermented for 72 h followed this sample in preference, being liked by 60–80% of the panelists, which was mostly associated with 1-hexanol and positive emotions such as joyful and adventurous. Samples with the least preference were COM at fermentation times of 0, 24, and 48 h, and were related to negative emotions such as disgusted, bored, aggressive, and worried.

Figure 3 shows the PREFMAP for taste, which was very different from that of odor. In this experience, the most preferred sample was COM, fermented for 72 h, which was also associated with positive traits such as free, satisfied, adventurous, and pleasant. In contrast, WIL, fermented for 48 h was now in the region with less preference, however, it was not associated with any negative adjectives.

## 4. Discussion

In Table 2, as much as ∆X and Q_p_ are concerned, there were no significant differences between strains; however, the growth rate (µ) and the production of lactic acid (∆P) differed. COM developed the highest µ out of the two strains while WIL outproduced COM in lactic acid synthesis. These differences may be explained by the niche specific characteristics from both strains since they were isolated from different origins, WIL, from Habanero pepper and COM, from tomato (*Solanum lycopersicum*). Growth rate (µ) may be affected by the metabolism of both strains, since COM was not originally isolated from Habanero pepper, it might focus on survival by the fastest growth rate, reaching a stationary phase sooner due to starvation, which is a common mechanism for lactic acid bacteria when carbon sources are scarce [21]. An inhibition by-product might also be taking place as the increase in the undissociated lactic acid concentration in the medium has previously been shown to decrease biomass production in other species of *L. plantarum* [22]. Furthermore, the isolation source mainly affects the metabolism of carbohydrates, for example, some strains can assimilate lactose, raffinose, sorbitol, and trehalose, while others do not [22,23]. This may be the case as lactose has been identified as one of the sugars present in different species of *Capsicum*, while it has not been found in some tomato samples, which might be the reason why WIL has a higher lactic acid production, while having a slower growth rate [24,25]. Another explanation might be the deviation of the metabolism of pyruvate, as *L. plantarum* is a heterofermentative lactic acid bacteria, which means that it is able to produce other metabolites as well as lactic acid during fermentation, and the main deviations from lactic acid can be the production of acetic acid or diacetyl [26]. However, the latter can be identified in the concentration of 2,3 butanedione during fermentation, which was only detected at one point for each strain and there was no significant difference between them. The change in volatile compounds through time for the alcohols 3,3 dimethyl-1-hexanol and 1-hexanol and the ester cis-3-hexenyl hexanoate might be due to fatty acid transformation through lipoxygenase pathways [27], which is possible as fatty acids such as palmitoleic and oleic acid have been identified in different species of *Capsicum* [23]. However, this might also be a niche specific adaptation as in previous works with *L. plantarum* LB-B1 isolated from milk, there was a significant production of 1-hexanol, which is the main product of COM, which, according to the provider, is mostly used for the fermentation of cheese even though it was previously isolated from tomato [28]. The increase in linalool can be explained by the ability of *L. plantarum* to produce glycosylases, able to elicit the bond between sugars and terpenes [29].

The influence of volatile compounds in odor preference might be related to the concentration. However, it is interesting that the most preferred sample, the fermentation of 48 h by WIL is characterized by its enhanced concentration of both 3,3 dimethyl-1-hexanol (from 1.40 to 3.19 µg mL^−1^), which increases as the peppers mature with peppery odor notes (characteristic of peppers), and trans-2-hexen-1-al (from 0.81 to 1.00 µg mL^−1^), characteristic of unripe peppers and related to green or leafy odor notes, and it is supposed to decrease as it matures for the synthesis of alcohols, however, it was enhanced by fermentation [30]. The second most preferred sample, the one fermented by COM for 72 h, was characterized by the production of 1-hexanol (from 2.10 to 6.01 µg mL^−1^), which might have been due to its detriment, as this compound is associated with off flavors because it has herbaceous odor notes [27].

Even though taste and odor are highly related, their results differed in this study as the preference for the samples was not distributed in the same way, possibly due to taste being influenced by other characteristics. For example, among the other nonvolatile compounds reported to affect flavor in pepper, it is important to mention two main groups: (1) sugars and (2) organic acids [31]. Lactic acid, the main organic acid in peppers, is associated with acidity during fermentation, which may have negatively influenced the 48 h fermented sample. Figure 1 shows the point where the lactic acid concentration reached its peak. It would be interesting in future work to evaluate how other organic acids reported in peppers (citric acid, malic acid, quinic acid, succinic acid, and fumaric acid) influence the sensory parameters such as preference and emotions.

## 5. Conclusions

Based on the results obtained, it can be concluded that the use of different strains in Habanero pepper fermentation had no significant effect on the biomass production or lactic acid production rate; however, it influenced the biomass production rate, where COM had the highest, and lactic acid production, where WIL was superior. The interaction between the strain and fermentation time presented a significant effect on the synthesis of volatile compounds such as cis-3-hexenyl hexanoate, 3,3 dimethyl-1-hexanol, 1-hexanol, and linalool while only time influenced the production of trans-2-hexen-1-al. WIL at 48 h of fermentation produced the highest concentration of 3,3 dimethyl-1-hexanol, trans-2-hexen-1-al, and linalool. Samples fermented with COM for 72 h shared the highest concentration for linalool and had the highest concentration of 1-hexanol and cis-3-hexenyl hexanoate. The most preferred sample for odor was the one fermented by WIL for 48 h, followed by COM for 72 h, and both were associated with positive traits: good, active, and happy for WIL and joyful and adventurous for COM, while unfermented samples were the least preferred and consequently associated with less positive traits: worried, aggressive, and bored. For taste, the most preferred sample was the use of COM to ferment Habanero pepper paste for 72 h as it was in the region of higher acceptance as well as being associated with the most positive traits: satisfied, free, pleasant, and adventurous, however, there were fewer preference regions in the preference map and not a single sample was associated with a negative trait. The most preferred compound for taste was the hexyl-3-methyl butanoate, the only one in the 60–80% preference region, associated with the emotion of nostalgia.

## Figures and Tables

**Figure 1 foods-11-03618-f001:**
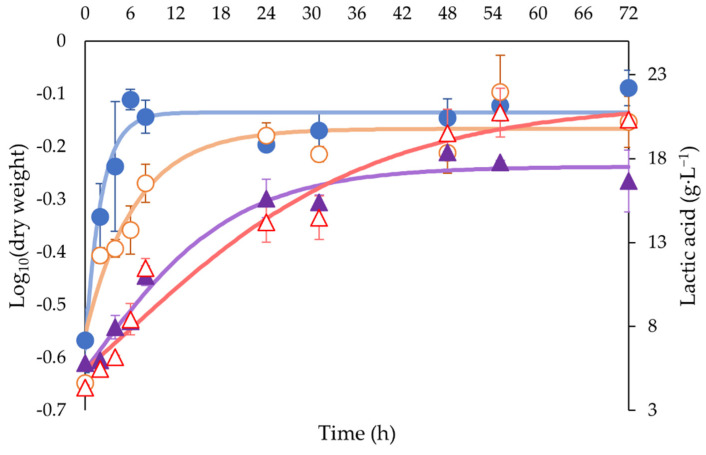
Log_10_ (dry weight) and lactic acid concentration for each strain during lactic acid fermentation. ● Dry weight for COM, ○ Dry weight for WIL, ▲ Lactic acid for COM, ∆ Lactic acid for WIL. Abbreviations: WIL, *L. plantarum* wild strain; COM, *L. plantarum* commercial strain.

**Figure 2 foods-11-03618-f002:**
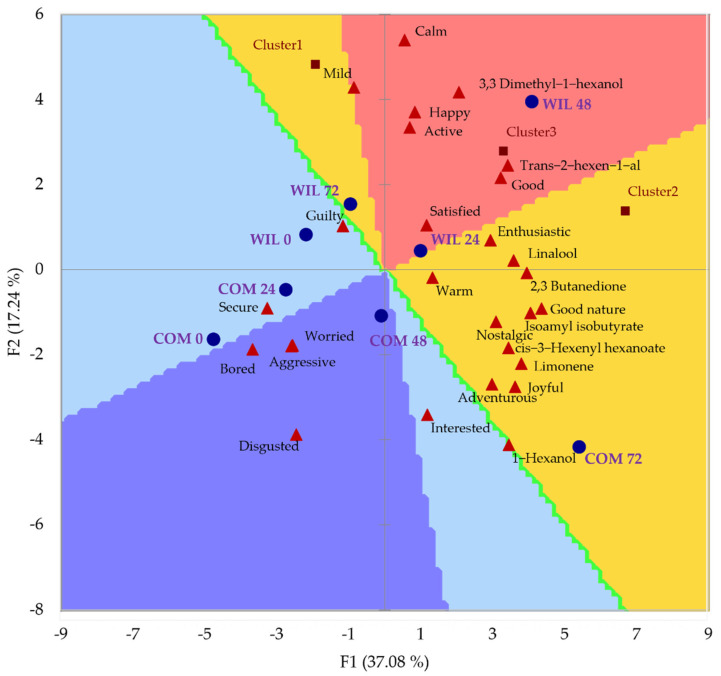
PREFMAP for the odor and PCA of volatile compounds and emotions for each strain and time during Habanero pepper fermentation. Symbology: ● Fermentation conditions ▲ Volatile compounds and emotions. Preference regions are represented with the following colors: ■ 80−100% preference ■ 60−80% preference ■ 40−60% preference ■ 20−40% preference ■ 0−20% preference. Abbreviations: PREFMAP, preference map; PCA, principal component analysis; COM 0, 0 h commercial *L. plantarum*; COM 24, 24 h commercial *L.* plantarum; COM 48, 48 h commercial *L. plantarum*; COM 72, 72 h commercial *L. plantarum*; WIL 0, 0 h wild *L. plantarum*; WIL 24, 24 h wild *L. plantarum*; WIL 48, 48 h wild *L. plantarum*; WIL 72, 72 h wild *L. plantarum*.

**Figure 3 foods-11-03618-f003:**
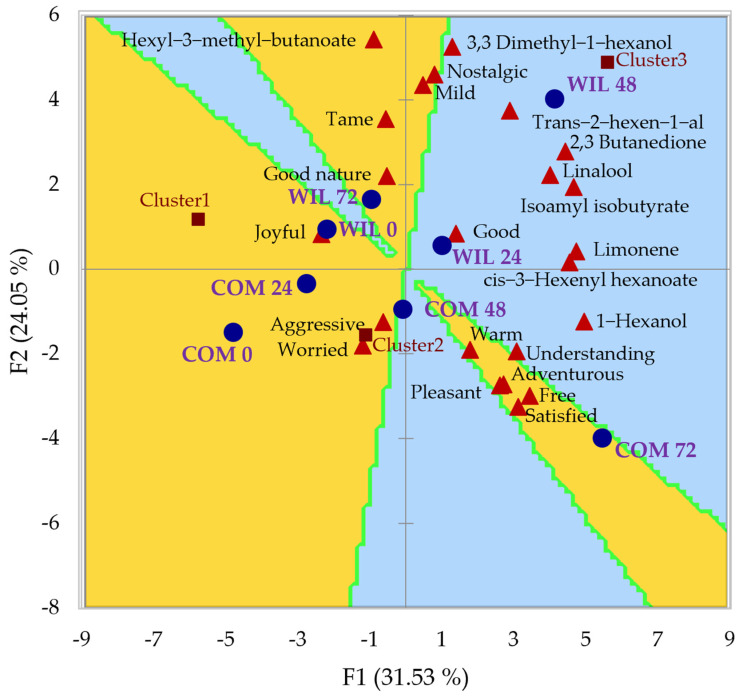
PREFMAP for taste and PCA of volatile compounds and emotions for each strain and time during habanero pepper fermentation. Symbology: ● Fermentation conditions ▲ Volatile compounds and emotions. Preference regions are represented with the following colors: ■ 80−100% preference ■ 60−80% preference ■ 40−60% preference ■ 20−40% preference ■ 0−20% preference. Abbreviations: PREFMAP, preference map; PCA, principal component analysis; COM 0, 0 h commercial *L. plantarum*; COM 24, 24 h commercial *L.* plantarum; COM 48, 48 h commercial *L. plantarum*; COM 72, 72 h commercial *L. plantarum*; WIL 0, 0 h wild *L. plantarum*; WIL 24, 24 h wild *L. plantarum*; WIL 48, 48 h wild *L. plantarum*; WIL 72, 72 h wild *L. plantarum* wild strain.

**Table 1 foods-11-03618-t001:** EsSense25 profile emotion terms [20].

Terms Used for Emotion Evaluation	
Active	Active
Joyful	Joyful
Adventurous	Adventurous
Loving	Loving
Aggressive	Aggressive
Mild	Mild
Bored	Bored
Nostalgic	Nostalgic
Calm	Calm
Pleasant	Pleasant
Disgusted	Disgusted
Satisfied	Satisfied
Enthusiastic	Enthusiastic

**Table 2 foods-11-03618-t002:** Growth and lactic acid production kinetic parameters in Habanero pepper fermentation.

Response Variable	Strain	
	COM	WIL
μ (h^−1^)	0.0680 ± 0.0025 ^b^	0.0336 ± 0.0074 ^a^
∆X (g L^−1^)	0.54 ± 0.03 ^a^	0.58 ± 0.06 ^a^
Q_p_ (g L^−1^ h^−1^)	0.7751 ± 0.1160 ^a^	0.8596 ± 0.1014 ^a^
∆P (g L^−1^)	9.91 ± 0.09 ^a^	10.81 ± 1.03 ^b^

Note: Results are reported as means. Values with a different lower case letter in the same column had statistical differences. Abbreviations: WIL, *L. plantarum* wild strain; COM, *L. plantarum* commercial strain; ∆X, highest dry weight production; µ, growth rate; ∆P, highest lactic acid concentration; Q_p_, lactic acid synthesis rate.

**Table 3 foods-11-03618-t003:** Concentrations of volatile compounds in the fermentation of Habanero pepper.

Strain	Time (h)	2,3 Butanedione (×10^2^ µg mL^−1^)	Limonene (µg mL^−1^)	Isoamyl isobutyrate (µg mL^−1^)	Trans-2- hexen-1-al (µg mL^−1^)	Hexyl-3-methyl-butanoate (µg mL^−1^)	1-Hexanol (µg mL^−1^)	Linalool (µg mL^−1^)	3,3 Dimethyl-1-hexanol (µg mL^−1^)	Cis-3-hexenyl hexanoate (µg mL^−1^)
COM	0	Nd	1.01 ± 0.00 ^a^	0.04 ± 0.00 ^a^	0.83 ± 0.01 ^a^	0.73 ± 0.00 ^a^	2.10 ± 0.09 ^a^	1.56 ± 0.04 ^a^	1.45 ± 0.03 ^a^	1.34 ± 0.53 ^a^
	24	Nd	1.01 ± 0.01 ^a^	0.05 ± 0.01 ^a^	0.82 ± 0.00 ^a^	0.73 ± 0.00 ^a^	2.46 ± 0.02 ^b^	1.80 ± 0.01 ^b^	1.65 ± 0.12 ^b^	2.38 ± 0.00 ^d^
	48	Nd	1.03 ± 0.01 ^a^	0.05 ± 0.00 ^a^	0.88 ± 0.02 ^a^	0.73 ± 0.00 ^a^	3.35 ± 0.67 ^d^	2.00 ± 0.12 ^c^	2.19 ± 0.24 ^c^	3.00 ± 0.22 ^f^
	72	8.16 ± 9.13 ^a^	1.07 ± 0.07 ^b^	0.11 ± 0.07 ^a^	0.89 ± 0.10 ^a^	0.73 ± 0.00 ^a^	6.01 ± 0.43 ^e^	2.14 ± 0.01 ^d^	1.85 ± 0.01 ^b^	3.40 ± 0.03 ^g^
WIL	0	Nd	1.03 ± 0.02 ^a^	0.06 ± 0.00 ^a^	0.81 ± 0.03 ^a^	0.73 ± 0.00 ^a^	2.32 ± 0.20 ^b^	1.65 ± 0.05 ^a^	1.40 ± 0.10 ^a^	1.46 ± 0.04 ^a^
	24	Nd	1.01 ± 0.01 ^a^	0.05 ± 0.00 ^a^	0.86 ± 0.01 ^a^	0.73 ± 0.00 ^a^	2.30 ± 0.03 ^b^	1.62 ± 0.01 ^a^	1.72 ± 0.06 ^b^	1.69 ± 0.04 ^b^
	48	8.11 ± 9.15 ^a^	1.04 ± 0.03 ^a^	0.09 ± 0.06 ^a^	1.00 ± 0.06 ^b^	0.74 ± 0.01 ^a^	2.95 ± 0.22 ^c^	2.14 ± 0.00 ^d^	3.19 ± 0.05 ^d^	2.75 ± 0.04 ^e^
	72	Nd	1.01 ± 0.00 ^a^	0.05 ± 0.01 ^a^	0.82 ± 0.00 ^a^	0.73 ± 0.00 ^a^	2.41 ± 0.10 ^b^	1.95 ± 0.04 ^c^	3.10 ± 0.70 ^d^	2.01 ± 0.01 ^c^

Note: Results are reported as means. Values with a different lower-case letter in the same column had statistical differences. Abbreviations: WIL, *L. plantarum* wild strain; COM, *L. plantarum* commercial strain; Nd, not detected.

**Table 4 foods-11-03618-t004:** Effect of the experimental factors (strain and time) during Habanero pepper fermentation over volatile compound concentration expressed as *p* values.

Factor	2, 3 Butanedione	Limonene	Isoamyl isobutyrate	Trans-2-hexen-1-al	Hexyl-3-methyl-butanoate	1-Hexanol	Linalool	3,3 Dimethyl-1-hexanol	Cis-3-hexenyl hexanoate
**A: Strain**	0.9955	0.7008	0.9937	0.4223	0.1606	0.0002 *	0.1458	0.0029 *	0.0007 *
**B: Time**	0.4202	0.4346	0.4995	0.0164 *	0.8433	0.0001 *	0.0000 *	0.0004 *	0.0000 *
**A*B**	0.1173	0.1976	0.2565	0.0666	0.5202	0.0001 *	0.0025 *	0.0201 *	0.0043 *

Note: An asterisk (*) indicates statistical differences. Interaction between factors A (strain) and B (time) is represented by A*B.

## Data Availability

The manuscript file contains all of the data.

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
