# Peer review of "Evaluation of the Volatile Composition and Sensory Behavior of Habanero Pepper during Lactic Acid Fermentation by L. plantarum"

_foods, 2022, doi:10.3390/foods11223618_

Round 1

Reviewer 1 Report

Research is good, but needs improvement to match  the standard of the journal

comments are attached,

Reviewer 2 Report

Additional remarks:

The quality and the scientific significance of this manuscript are acceptable, I found only some problematic parts, as you can see below.

Detailed review:

Suggested title:

Evaluation of the volatile composition and sensory properties of Habanero pepper during fermentation by L. plantarum

Line 65: what were two strains? Please add: “wild” and “commercial” phrases!

Line 83: how counted the cells?

Lines 94-106: rather put into the statistical Analysis subsection!

Line 110: how counted the cells? Need reference!

Line 167: EsSense25 – reference?

Table 2: lower case letters: “a” indicate the smallest value, while “b” is the highest value (ΔP-row)!
